# Dopaminergic Gene Dosage Reveals Distinct Biological Partitions between Autism and Developmental Delay as Revealed by Complex Network Analysis and Machine Learning Approaches

**DOI:** 10.3390/jpm12101579

**Published:** 2022-09-25

**Authors:** André Santos, Francisco Caramelo, Joana Barbosa Melo, Miguel Castelo-Branco

**Affiliations:** 1Coimbra Institute for Biomedical Imaging and Translational Research (CIBIT), ICNAS, Faculty of Medicine, University of Coimbra, 3000-548 Coimbra, Portugal; 2CIBB, iCBR, Faculty of Medicine, University of Coimbra, 3000-548 Coimbra, Portugal

**Keywords:** Autism Spectrum Disorder, genetics, dopamine, complex networks, machine learning

## Abstract

The neurobiological mechanisms underlying Autism Spectrum Disorders (ASD) remains controversial. One factor contributing to this debate is the phenotypic heterogeneity observed in ASD, which suggests that multiple system disruptions may contribute to diverse patterns of impairment which have been reported between and within study samples. Here, we used SFARI data to address genetic imbalances affecting the dopaminergic system. Using complex network analysis, we investigated the relations between phenotypic profiles, gene dosage and gene ontology (GO) terms related to dopaminergic neurotransmission from a polygenic point-of-view. We observed that the degree of distribution of the networks matched a power-law distribution characterized by the presence of hubs, gene or GO nodes with a large number of interactions. Furthermore, we identified interesting patterns related to subnetworks of genes and GO terms, which suggested applicability to separation of clinical clusters (Developmental Delay (DD) versus ASD). This has the potential to improve our understanding of genetic variability issues and has implications for diagnostic categorization. In ASD, we identified the separability of four key dopaminergic mechanisms disrupted with regard to receptor binding, synaptic physiology and neural differentiation, each belonging to particular subgroups of ASD participants, whereas in DD a more unitary biological pattern was found. Finally, network analysis was fed into a machine learning binary classification framework to differentiate between the diagnosis of ASD and DD. Subsets of 1846 participants were used to train a Random Forest algorithm. Our best classifier achieved, on average, a diagnosis-predicting accuracy of 85.18% (sd 1.11%) on the test samples of 790 participants using 117 genes. The achieved accuracy surpassed results using genetic data and closely matched imaging approaches addressing binary diagnostic classification. Importantly, we observed a similar prediction accuracy when the classifier uses only 62 GO features. This result further corroborates the complex network analysis approach, suggesting that different genetic causes might converge to the dysregulation of the same set of biological mechanisms, leading to a similar disease phenotype. This new biology-driven ontological framework yields a less variable and more compact domain-related set of features with potential mechanistic generalization. The proposed network analysis, allowing for the determination of a clearcut biological distinction between ASD and DD (the latter presenting much lower modularity and heterogeneity), is amenable to machine learning approaches and provides an interesting avenue of research for the future.

## 1. Introduction

### 1.1. The Putative Role of Dopaminergic Signalling in ASD

Autism Spectrum Disorder (ASD) is a neurodevelopmental disorder characterized by social and communication impairments and restrictive and repetitive behaviors and interests, yet its etiology and neurobiological mechanisms are still poorly understood [1]. The attribution of the diagnosis of ASD is assessed accordingly with the guidelines present in the latest version of DSM (DSM-5). It admits only one macro-clinical condition, ASD, and allows clinicians to specify the severity of different symptoms (which differs from previous versions of DSM where different diagnostic profiles existed to differentiate diverse symptom severity).

In the last years, many genetic variations have been associated with ASD; however, in most cases, these patterns were only verified in small-size samples leading to a lack of clarity [2,3,4,5]. The large range of different individual genetic manifestations and still-unknown gene–environment interactions further compounds this problem [6,7]. Research on putative causal aspects that relate to the different symptoms of ASD remains a priority. The role of dopaminergic neurotransmission and its involvement in the causal pathway of ASD deserves attention in this regard [8,9,10]. For example, disruptions in the nigrostriatal pathway, now known to go beyond the classical link to motor functions, have been proposed as a potential contributing cause of repetitive behaviors in ASD whereas disturbances in the mesolimbic and mesocortical pathways, involved in reward- and cognitive-related functions, may lead to affective and social cognitive impairments [11,12].

### 1.2. Addressing Complex Diseases with Complex Network Approaches

Complex diseases are often caused by a combination of many intrinsic and extrinsic factors. Thus, this definition also implies that the disease’s cause can rarely be explained by a single perturbation. Most neurodevelopmental diseases, such as autism, schizophrenia and intellectual disability fall under this definition. Here we will focus on the genetic causes.

The multiple genetic causes behind a complex disease can be hard to identify given the difficulty in isolating the individual effects of single genetic alterations. Most of the time these individual effects might be small and rare, further increasing the difficulty in correctly identifying their effects. For example, it is recognized that autism is highly heritable, but, given the role of rare genetic variations, causality in this respect remains a conundrum [13,14].

Another aspect increasing the difficulty of studying complex diseases comes from heterogeneity. Typically, the same diagnostic category is attributed to a spectrum of similar phenotypes, yet it is unlikely to find two patients with the same diagnosis and identical phenotypes. This explains why this condition has been renamed Autism Spectrum Disorder, to encompass a set of conditions related to impairments in social interaction and communication and stereotyped, repetitive behaviors [13]. However, the spectrum admits both individuals totally dependent upon life support and individuals that can live almost independently.

Therefore, this multifactorial scenario poses a big scientific challenge. Systems biology approaches might be suitable to address this problem through analysis of complex networks mapping interactions between genes, proteins, molecules and disease phenotypes. For example, in a disease network it is possible to observe several genetic alterations linked to the same pathophysiological process where a perturbation in one gene can be propagated and affect other genes throughout network interactions. This type of approach has often led to the observation that different genetic backgrounds lead to similar phenotypes (phenocopies) in line with the idea that these causes disturb the same biological mechanism rather than being disconnected or acting in an isolated manner [15].

According to this view, molecular or cellular components belonging to particular functional modules act in concert to perform essential biological tasks, and such interplay can lead to abnormal interactions that are characteristic of complex diseases. These interactions can be represented as networks where nodes denote genes, gene products, biological functions and/or disease categories, and links denote the interactions between them.

A network of biological interactions has characteristic topological properties [15]. One of these properties, shared by almost every biological network, is the scale-free property [16]. Scale-free networks have a node degree distribution similar to a power-law distribution meaning that a small number of nodes have a high number of connections while the majority interacts only with a few neighbors. The highly connected nodes are called hubs, and are of particular interest, because they frequently play a crucial role in a biological mechanism [16]. Extending the analysis to the hub’s neighborhood allows for the to identification of patterns of connectivity or groups of nodes underlying the logic of particular biological functions [17,18].

### 1.3. Prediction of ASD Diagnosis with Machine Learning

Machine Learning (ML) approaches have proved useful in many health-related classification tasks. One of the strongest examples of this is the field of radiomics [19]. With respect to ASD, ML classification has been applied in several studies to predict diagnosis; however, prediction results are still limited.

A 2019 meta-analysis on the subject [20] observed a discrepancy in the diagnostic-prediction accuracy between 60% and 98% across 57 different studies. In this analysis the authors conducted a literature search on all studies that attempted to predict ASD clinical diagnosis status either cross-sectionally or longitudinally on the basis of biology, cognition and/or behavior, implementing a case-control design with only out-of-sample predictions (i.e., a predictive model was trained on one part of the data and tested on another). The reason given for such disparity was attributed to several aspects; for example, bias in the sample, small sample sizes, usage of different validation methods, the heterogeneity of ASD and data quality.

Another key factor that could be observed in the meta-analysis is related to the type of data and scientific domains where the classification tasks were applied. We detected that of the 57 studies, only one focused on genetic features to predict ASD diagnosis. In this particular study [21], data from 487 ASD patients and 455 healthy individuals were used to build an ML classifier. It relied on single-nucleotide polymorphism (SNP) data to predict the diagnosis. The accuracy, sensitivity and specificity achieved by their model were 73.67%, 82.75% and 63.95%, respectively.

Here, we decided to focus on the distinction between ASD and DD as an important intermediate step, because of the relevance of differential diagnosis between clinical categories, and most importantly, because these categories might provide a clear-cut biological distinction. We present an ML approach to predict diagnosis with genetic Copy Number Variations (CNV), aiming to address the aforementioned issues and propose a novel framework using genetic data to predict the clinical category.

### 1.4. Study Approach and Aims

In order to contribute to the discussion on the impact of the dopaminergic system in ASD, and its biological properties, we studied several dopaminergic features coded in the CNV gene content of ASD carriers [22]. A CNV involves unbalanced rearrangements that increase or decrease DNA content and that can affect several nucleotides in a chromosomic region. This type of alteration can occur across the genome and can be detected using CMA (Chromosomal Microarray Analysis) technology. Genome-wide CMA is a first-tier test for most neurodevelopmental disorders [23,24] and the CNVs detected through it are classified into five categories according to the guidelines of the American College of Medical Genetics and Genomics (ACMG) and Clinical Genome Resource (ClinGen) [25]. These categories represent different levels of a CNV being associated with a disease and include the following: (1) Pathogenic, (2) Likely pathogenic, (3) Uncertain Significance, (4) Likely Benign and (5) Benign.

However, given the characteristics of complex diseases, we wonder about the impact of multiple CNVs, their interplay at different risk-levels and the potential of CMA technologies to help studying this problem. Hence, here we study participants with multiple CNVs where dopamine-related genes were in duplicated or deleted chromosomic regions.

Firstly, we used QuickGO [26] to identify a set of gene ontology (GO) [27,28] terms related to the dopaminergic system. Next, from the SFARI Gene CNV Module [29] (sfari.org/resource/sfari-gene/, accessed on 17 September 2017), we selected participants having CNVs matching genes of interest based on the previously defined GO terms.

To address the genetic variance, frequency and heterogeneity in ASD, we used complex network analysis [30]. This approach also allowed us to model the data accordingly to a functional polygenic view [31,32] through maps of interactions between participant diagnosis, genetic alterations and affected biological mechanisms.

In the end, we transformed the network’s information into vectors suited for an ML classification problem. We used Random Decision Forests [33] to predict the participants’ differential diagnosis when presented with dopamine-related features extracted from gene alterations observed in participants.

## 2. Materials and Methods

### 2.1. Data Source and Participant Selection

With the QuickGO API we created a set of 110 GO terms related to dopaminergic neurotransmission aspects that were used to identify a set of 125 genes from CNVs in the public dataset curated by the SFARI Gene CNV module. The genes within each CNV region were identified using the BioMart API [34,35]. For this study, we admitted participants within the SFARI Gene CNV module with duplications or deletions in the genes identified previously. We only included in the analysis participants with a single clinical diagnosis. Overall, we selected 1318 (Male: 699; Female: 134; Not Specified: 485) participants with diagnosis of ASD and 1327 (Male: 81; Female: 67; Not Specified: 1179) participants with diagnosis of developmental delay (DD). The raw data used to start this study as well as the data used to construct the networks and to perform the machine learning approach are available in this public repository: https://dataverse.harvard.edu/dataset.xhtml?persistentId=doi:10.7910/DVN/HO1JLJ (published on 8 March 2022) [36].

### 2.2. Building Networks of Participant’s Genomic Features and Diagnostics

We continue our approach by building a network using participants and gene alterations as nodes [37]. In this network (Gene Dosage Network) each participant was linked to a gene if a duplication or a deletion was present. Then, a network using GO terms and participants as nodes (GO Network) was built by replacing each gene with the associated GO terms. Lastly, we joined all information into a single network, consisting of links between genes, GO terms and participants (Gene Dosage–GO Network). All networks were built using NetworkX package (version 2.4, created by Aric Hagberg, Dan Schult, and Pieter Swart, Los Alamos, USA) [38] and layout was performed with Gephi software (version 0.9.2, created by Mathieu Bastian, Sebastien Heymann and Mathieu Jacomy, Paris, France) [39] to show a spatial arrangement of the nodes and their links. Node degree (the number of links attached to a node) was used to size nodes and colors were used to mark different types of nodes. The Fruchterman–Reingold (FR) algorithm [40] organized the nodes using a gravity approach where the higher the degree of a node, the stronger the force by which it attracts the linked nodes and pushes the unlinked ones. In order to enable visual comparisons, the parameters used on the FR algorithm—namely, the area used to display the nodes, the gravity force and the speed at which changes occurred until the stabilization of the algorithm—were set equal for all networks.

### 2.3. Network Analysis Methods

From the network topologies we extracted centrality measures [41,42], such as the number of nodes, the number of edges (links), the average degree and the network diameter. In graph theory, the degree of a node identifies the number of connections/links that a node has, whereas the degree distribution gives the probability of finding a node in a network for a given degree. The degree distribution of each network was compared to the Poisson curve and the power-law curve. The Poisson curve represents a distribution of the node degree, where most of the nodes have nearly the same degree with small deviations from the average; this type of curve has a bell-like shape. In this scenario, the existence of a node with higher degree than the average of the network degree is unlikely. Such a node will be an outlier. Contrarily, the power-law curve predicts the existence of fewer nodes with higher degree, named hubs (outliers in a universe of nodes characterized mainly by lower degree nodes). This approach helped to understand if the networks were closer to the Random Network model, described by the Poisson curve, or to the Scale-Free Network model, described by the power-law curve [43].

Next, we analyzed the hubs of each network in terms of their degree and the average degree of their neighbors. The former provides information about the number of participants linked to a particular disrupted gene or GO term. The latter gives information about the average of participant interactions with other genetic imbalances beyond the hub [44]. These measures help to understand the importance of the hub in the network structure. For example, a hub with high degree and average neighbor degree equal to one represents a hub-and-spoke pattern, meaning that all its neighbors have on average only one link; that link could only be the link to the hub. In this scenario, the hub is a central piece to keeping the network connected; removing it will disconnect all the nodes linked to it, breaking the structure of the network and consequently its scientific meaning. Another possible scenario occurs when the hub has an average neighbor degree higher than one. This case informs about participants linked to more than one gene or GO term where the hub is a central part of a subnetwork. The study of these subnetworks could help identifying groups of genes or biological mechanisms shared by participants. Therefore, we conclude our network analysis addressing this type of interaction. Using participants’ generalized similarities, we tried to identify groups of genes and GO terms strongly shared among participants within the same diagnostic category. This type of knowledge helps not only with comparing differences between clusters of biological mechanisms related to ASD and DD diagnostics, but also allows us to collate differences between participants within the same diagnostic category [45].

### 2.4. Applying Machine Learning in Differential Diagnosis

We transformed the features of the created networks in data for an ML classification problem [46,47]. Figure 1 provides an overview of the ML approach. The target classification variable was the clinical diagnosis of participants, and to predict it we used features extracted from the network topology. We measured the importance of each non-participant node by dividing the node degree (the number of other nodes attached to it) by the total number of links in the network the node belongs to; for each participant, if it was linked to a particular node, we use its node importance multiplied by the type of the participant gene dosage (Figure 2). For example, in the gene network, each gene was a feature. The value for each feature is given by the node degree over the total number of the network links, which we denoted as network node importance, multiplied by a factor of 1 or −1 representing the participant CNV type; a duplication or deletion, respectively. The gene ontology network was framed in the same way. However, for the last network we constructed the features differently. For a given GO term in the network, its value was given by the importance of the gene node it was linked to plus its own node importance, multiplied by the participant CNV type (Figure 2). We obtained three datasets relying on different types of data: (1) gene dosage, (2) GO and (3) combined gene dosage and GO data.

The SciKit-Learn package (version 0.21.3, first released by Fabian Pedregosa, Gael Varoquaux, Alexandre Gramfort and Vincent Michel, Rocquencourt, France) [48] was used to train and to test Random Forest classifiers in order to predict the participants’ diagnosis based on network features and on the participants’ type of gene dosage, duplication or deletion. We started by randomly splitting the participants into a training and a test set, with a 70/30 ratio, under an equal distribution of classes. In the training set we identified the features with more discriminant power using a wrapper methodology with a threshold of 1×10^−3^ for feature importance [49] which were then used to train the classifier. The test set was validated on the previously trained algorithm and the confusion matrix was recorded. This process was repeated 100 times for each type of dataset. At the end, we calculate the mean and the standard deviation of several metrics, such as the accuracy, the precision and the recall, from the confusion matrix obtained in each repetition [50].

## 3. Results

Data used in this work come from SFARI Gene CNV Module which gathers CNV data related to ASD from several studies. In Table 1 the number of participants discriminated by diagnostic categories are listed. The listed participants have CNVs where dopamine-coding genes were present. About 94% were diagnosed with DD (47.14%) or ASD (46.82%), providing an adequate balance for the analysis, whereas the remaining 6% were distributed over thirteen other diagnoses, for example, schizophrenia (1.78%) and intellectual disability (1.67%). Our analysis proceeded with participants having the diagnosis of ASD or DD due to the differences shown in the frequency of the diagnoses, which led us to focus on this type of differential diagnosis and the underlying neurobiology. Given the number of participants without sex specification we decided not to do a split-sex analysis.

### 3.1. Macroscopic Network Features

The methodology used to construct the networks allowed us to highlight macroscopic differences between networks (Figure 3, Figure 4 and Figure 5). For example, in the Gene Dosage Network, related to dopaminergic neurotransmission (Figure 3), nodes with larger size are found near the limits of the figure and distant from each other, whereas in the GO Network (Figure 4) a distinct and reorganized pattern is observed: larger nodes appear at the center of the figure and closer to each other. This is likely related to the number of links each larger node has attached and with the number of links in its vicinity which impacts the results of the Fruchterman–Reingold algorithm. This algorithm relies heavily on gravity concepts to display the nodes in a network (consult Section 2.2. for technical details).

In the Gene Network (Figure 3) participants were linked either to a single gene or instead to a set of genes, so each larger gene node will pull their linked participants and push the other (unlinked) nodes way. In Figure 4 (GO Network) we swap the genes nodes by their corresponding GO terms and the links organization between the two types of nodes (participants and GO terms) changed completely. This resulted in an interesting and revealing structural alteration of the network (in comparison to the Gene Dosage Network) showing that many participants share dysregulations in sets of the same dopaminergic domains although prevenient from different genes.

Furthermore, we observed that the percentage of gene nodes (7.44%) in the Gene Dosage Network was higher than the sum of the percentage of the different GO terms nodes (total = 2.73%; biological processes: 1.88%; molecular functions: 0.81%; the cellular component: 0.04%). This property might be helpful when analyzing problems with high genetic variance and low occurrence.

Importantly, in this work we were aware that by only looking at one of the types of networks we might be missing the information contained in the other. Thus, to overcome this problem, we built the Gene Dosage–GO Network (Figure 5) by linking participants, genes and GO terms.

### 3.2. Network Analysis

Analyses of centrality measures, degree distribution and hubs helped to confirm the previously observed patterns, and to identify new relations between dopaminergic neurotransmission features and the underlying diagnosis of the participants under different contexts. These may be difficult to detect if one relies only on simple descriptive analysis of visual patterns. For example, simply counting the number of nodes and links of each network is quite cumbersome. Table 2 presents the centrality measures of each network. Considering N (number of nodes) and L (number of links) we identify that the Gene Network has higher N and lower L than the GO Network. This decreases the <k> (average degree) and density properties of the Gene Network. Compare to the other two networks, we observe that the Gene Network is the most poorly connected, in line with the notion that it is more limited in providing direct biological information.

Moreover, when we analyze the diameter of the networks, we observe that it is decreased ~2.3 times from the Gene Network to the GO Network; this result is in agreement with our initial prediction that, in the Gene Network, nodes appear more distant than in the GO Network, which has thereby larger biological meaning. Indeed, swapping genes by their linked GO terms resulted in seven degrees of separation, meaning that a node in this network can only be at a maximum path distance (the maximum number of nodes linked between two nodes of interest) of seven nodes from another. Thus, this transition enhanced a property of complex networks known as small-worlds [51,52,53], which in this particular case, forecasts the existence of more communities (smaller groups of nodes strongly tied in a larger networks) in the GO Network than in the Gene Network. The study of communities has strong scientific value since it allows researchers to identify key biological mechanisms disrupted in groups of participants; in turn, this can be used, for example, to partition in a meaningful way the ASD heterogeneity derived from genetic factors.

Next we present the results of contrasting the networks’ degree distribution with the Poisson distribution and the power-law distribution [54,55] (Figure 6 and Figure 7). This allows us to understand if networks were closer to a Random Network model, which expects the inexistence of nodes with much higher degree than the average network degree, or to a Scale-Free Network model, which expects a higher number of nodes with lower degree and the existence of fewer critical nodes with extremely high degree [43]. Here, we observed a long-tail behavior in both networks which favors the proximity to a Scale-Free model; in other words, in our networks there were nodes with larger degree in a such way that they could be reliably labeled as hubs. These hubs are central parts in the network structure and, therefore, their study is of particular interest to uncover the biologic meaning behind the network.

#### 3.2.1. Hubs and Neighborhood Analysis

In Table 3, for each network we report the nodes with the highest degree and the average degree of their neighbors. In the Gene Network, the major gene hub (ENSG00000102882, *MAPK3* gene) is linked to 463 participants and each one of these participants had on average 1.6 links, meaning that most of these participants are only linked to this hub. Thus, this hub is extremely important to uphold the network as one big interconnecting component. It represents a hub-and-spoke pattern, so removing it will disassemble the network, producing several isolated participant nodes, resulting in loss of information. In other words, the imbalances in the *MAPK3* gene are a major aspect for ASD and DD participants. On the other hand, when we look at the last hub (ENSG00000050628)—related to *PTGER3*—listed for the Gene Network, we observe that the node is linked to 339 participants and has an average neighbor degree of 11.8. In this case, each participant linked to this hub was also linked to another 10 genes on average, and removing it from the network will not break its connectivity [56,57]. However, this gene is also very important for ASD and DD analysis since it identifies a subnetwork of disrupted genes that could be under the same biological domain.

Comparing the hubs of the two networks, we verify that the GO Network hubs were, as expected, linked to more participants and that these participants share several links with other GO terms. For example, the major hub of the GO Network had more than twice the links of the major hub of the Gene Network, and all of its participants were also linked on average to another six GO terms. Again, Table 3 supports the prior analysis that, in the GO Network, the GO nodes seemed to be more connected and closer to the center due to the existence of more links between participants and GO terms than participants and gene nodes in the Gene Dosage Network.

Additionally, we observed that the GO term hubs were related with the biological processes of the gene hubs: (1) Cellular response to dopamine is a biological process of *MAPK3*, (2) Dopamine metabolic processes are related to *COMT* and (3) The last GO term hub is a synonym of the phospholipase C-activating dopamine-receptor-signaling pathway which, in turn, is a term attributed to the *PTGER3* gene. Thus, the information in the Gene Network remains across the GO Network.

Moreover, since GO hubs had more links than gene hubs, it allowed us to aggregate other participants with the same disrupted biological mechanism but stemming from other genetic sources of lower degree. The average neighbor degree suggests the existence of several subnetworks of biological mechanisms shared between groups of participants which could be related to distinct dopaminergic domains. Consequently, these resulted in GO hubs with higher degrees, and in higher neighbor degree, increasing the network’s information value. It is also a more-robust and less-vulnerable network; in other words, its risk of being disassembled due to a node deletion decreased due to the increase of links between nodes and node neighbors [58].

#### 3.2.2. Generalized Similarity within ASD and DD

Here we analyzed similar participants by grouping their shared genes and GO terms in a network. The analysis was performed individually for each type of diagnosis not only to spot biological differences between diagnostic categories but also to investigate differences within participants under the same diagnosis. Figure 8 and Figure 9 depict groups of genes shared by similar participants with ASD or DD, respectively. Each cluster of genes represents a set of participants sharing similar genetic alterations, and the colors represent the individual gene’s strength of ASD risk according to the SFARI Gene Score Module assessment.

In the ASD Gene Network we identified 163 genes distributed across 21 communities (subnetworks within a network). These groups are characterized by distinct genetic signatures of different groups of participants with ASD. Moreover, the network is relatively disassembled; it is composed of 17 connected components isolated from each other, i.e., without links between them. These components represent 17 unique groups of participants each with their own unique subset of genes. Within the network were identified 3 genes with high evidence of ASD risk, 2 strong candidate genes and 16 suggestive candidates. However, 142 genes had no information about the respective individual risk. The DD Gene Network structure is analogous to the ASD Gene Network. It counts 64 different genes distributed in 11 communities. Again, the network is disassembled in 10 connected components, showing genetic variability within the DD diagnostic category. One difference is the absence of genes classified as high confidence of ASD risk.

Regarding the similarities between participants and their GO terms, we present Figure 10 and Figure 11 for ASD and DD types of diagnosis, respectively. The network nodes were colored according to the modularity, which allowed for the detection of communities within a network by evaluating the way nodes are linked between each other. This helped to spot imbalanced clusters of biological mechanisms that characterize different participant subsets. The ASD GO Network is formed by 41 nodes and nine communities. Here exists only one connected component, and this has two important implications. The first is related to the genetic variability observed in Figure 8 for ASD genes. These results show that different imbalanced genes are indeed disrupting the same biological mechanisms. The second is related to the heterogeneity of ASD. The connected component and its nine communities suggest that some ASD participants could share signatures from different communities. In four of these communities, we identified a principal biologic mechanism: (1) orange nodes are related to the binding in dopamine receptors, (2) pink nodes are related to dopamine-metabolic processes, (3) green represent mechanisms that regulate the differentiation of dopaminergic neurons and (4) red nodes represent dopaminergic synapse biology. Moreover, there is a proximity between communities of similar domains. For example, the orange community (dopamine-receptor binding) is neighbor to the pink community (dopamine-metabolic process), whereas the green community (regulation of dopaminergic neuron differentiation) is close to the red community (dopaminergic synapse). Thus, the results show that different combinations and levels of these factors may explain different subsets of ASD participants.

On the other hand, the DD Network is constituted by 13 GO term nodes spread by two communities. As in the previous network, all its nodes are connected into one single component; however, the modularity level is very low (0.097), which results in one larger community. This community is formed by several key dopaminergic concepts, such as dopaminergic synapse, regulation of dopamine-metabolic process, dopaminergic differentiation and regulation of dopamine secretion. Thus, this result suggests that the genetic imbalances of most DD participants share dysregulations in all these biologic mechanisms simultaneously. It differentiates from the ASD diagnosis because the genetic variability was higher, and within subsets of ASD participants, only some of these aspects were dysregulated. Thus, this indicator has potential for tasks addressing the differentiation of ASD from DD diagnoses and also in differentiating within ASD.

### 3.3. Statistical Classification between ASD and DD

In the previous analyses, we observed that some nodes had more links than others had, and assumed that they had different importance. Therefore, we used this to build features for a classification problem (differential diagnosis between ASD and DD). The results of the ML approach are presented in Table 4 and revealed an average accuracy at predicting the differential diagnosis of the participants in the test sets of 85.18% (±1.11%), 83.22% (±1.09%) and 85.13% (±1.06%) for Gene Dosage, GO and Gene Dosage–GO datasets, respectively.

To understand the types of errors in the decisions that lead to the test accuracy results, we verified the confusion matrix and analyzed both the false positive and false negative predictions. For example, we found that in the Gene Dosage test sets (which contained 395 participants diagnosed with ASD and 395 diagnosed with DD) on average 25 (sd 5) ASD participants were misclassified with DD diagnosis and 92 (sd 8) DD participants were misclassified with ASD. This pattern is present in the other datasets used in the ML approach, and influenced the recall, precision and ultimately the F1 score obtained.

Overall, the classifiers’ ability to identify participants with ASD diagnosis was higher than the ability to identify DD participants; in other words, the precision was higher in ASD diagnosis compared to DD diagnosis independently of the dataset used. However, the confidence when the classifier marked a participant with DD was higher than the confidence when the classifier marked a participant with ASD; to put this in another way, recall was higher in DD diagnosis compared to ASD diagnosis for each dataset used. This resulted in higher F1 scores for DD diagnosis compared to ASD diagnosis.

#### Impact of Feature-Type in Classifier’s Performance

Finally, we will focus on the number of features used to train and test the classifiers recorded in Table 4. On average, 117 (sd 4) genes were needed to train a classifier with gene dosage data, whereas 62 (sd 2) GO terms were used for the same purpose with a drop in the test accuracy of approximately ~2%; in other words, with nearly half of the features it was possible to train a classifier with almost the same performance by swapping the genes with its corresponding GO terms. This fact may impact the classifier’s ability of generalization [59,60] in ML approaches applied to ASD due to the low effective contribution of individual genes and high genetic variability. However, we were aware that relying only on one type of data was not an optimal solution. As demonstrated earlier, either genes or GO terms had different importance. In order to express the GO term’s importance as a function of the gene’s importance we built the Gene Dosage–GO dataset. Surprisingly, by combining gene and GO features, we were able to decrease the required number of features to train the classifier to a minimum of 59 (sd 2) features and maintain the classifier-performance metrics as similar to the other two individual approaches (Gene dataset and GO dataset).

## 4. Discussion

The present study provides novel insights on the underlying neurobiology of ASD and DD which concerns the nature of dopamine gene-dosage effects. We highlighted key biological mechanisms underlying observed phenotypes by mapping relations with several dosage-imbalanced dopamine-related genes. Furthermore, we showed that they may consistently be used when applied to a clinical diagnosis problem of classification based on polygenetic profiles. Our approach is novel and allowed for addressing issues, such as genetic variability, which has been largely pointed out by previous studies.

One of the main issues when studying complex and heterogeneous diseases, namely ASD, is related to the need for a large sample size. In this work we used a sample size of 2636 participants (Table 1), whereas in other studies aiming for diagnostic classification in ASD the maximum number of samples used was around 1000 participants [21,61,62,63,64,65,66]. This large number of participants allowed us to frame our problem and test our methodology around the genetic variability associated with such complex diseases. However, with our methodology it was not possible to apply a sex analysis, given the number of participants with unspecified sex would have reduced our sample to 981 participants. The imbalance of the number of participants with specified sex between ASD (699) and DD (148) was another motive to not proceed with sex analysis with this sample. For example, a cut in the number of participants will have resulted in a drop of genes and gene ontology terms studied. This action would probably impact the ML results since the number of samples and the number of features available will differ between ML analyses. Consequently, it would be difficult to specify if the difference observed was a result of splitting between sex or a result from a cut in the sample size and number of features. Thus, we made the choice to make the analysis with the largest sample we could find without a split between sex.

We used a polygenic approach to map the participant relations with multiple genes which encode several aspects of dopaminergic biology and function (Figure 3). We followed the systems biology tenet that it is important to go beyond simple gene dosage concepts such as gene duplication or deletion. Using GO terms, we explored and mapped the relations between participants and their genetic signatures, linking participants with similar disrupted biological processes and molecular functions occurring in the dopaminergic synapse (cellular component) (Figure 4). The Gene and GO Network description rendered quite obvious the need to use appropriate tools for the analysis of complex diseases. This was achieved using complex network analysis where we demonstrated some unique biological characteristics underlying ADS and DD categories.

Firstly, we observed the presence of critical hubs in both the Gene and the GO Network (Figure 6 and Figure 7). It is a unique feature of scale-free networks and, in this case, indicates that some genes and biological mechanisms appear to be most frequently disrupted among the participants. In the top of this hub’s ranks we found *MAPK3*, *COMT* and *PTGER3* genes (Table 3). A distinct aspect between them is the number of links in their vicinity. Participants linked to *MAPK3* and *COMT* genetic alterations do not share links with any other genes, whereas participants linked to *PTGER3* were part of a subnetwork of genetic alterations. Thus, there were participants in whom the disruptions in dopamine pathways were encoded only in one gene and others where it was encoded across a subset of genes.

Additionally, each of the GO terms identified as hubs were related to distinct biological functions encoded by each of the hub genes: (1) The cellular (synaptic) response to dopamine depends on information coded in *MAPK3* which plays a critical role in diverse biological functions involving the *MAPK*/*ERK* cascade, particularly during neurodevelopmental stages [67]. (2) The dopamine-metabolization processes are partly dependent on *COMT* genetic coding since its expression is essential to the catalyzation of neurotransmitters of the catecholamine family, including dopamine, epinephrine and norepinephrine. The modulation of this pathway is relevant in neurodegenerative disorders [68], and in ASD its regulation is associated with abnormal dopamine levels, abnormal brain activity and increased severity of autistic behaviors, although some evidence remains controversial [69]. (3) The phospholipase C-activating dopamine-receptor-signaling pathway is related to the *PTGER3* gene, as the protein encoded by this gene is a member of the G-protein-coupled receptor family which is relevant across pivotal metabolic domains [70] relevant to many human diseases, including ASD [71].

Moreover, we note a distinct pattern in the GO hubs; it always had more participant links and always had several interactions in their vicinity. This suggests that these are more informative and that genetic imbalances from various gene sources are also targeting these biologic concepts. However, they are not spotted under the gene network. In sum, definable subnetworks of biological mechanisms where frequently disrupted in ASD and DD.

Secondly, we proceed to detect subnetworks of genes and GO terms shared by joint sets of similar participants. In the ASD Gene Network and DD Gene Network (Figure 8 and Figure 9) we found several clusters of genes exclusive within subgroups of ASD and DD participants, respectively. Thus, there were no connections between clusters, which is in accordance with the distinct genetic variability present in these complex diseases. Additionally, we found some genes that had a different individual strength of ASD individual risk according to SFARI Gene Score Module assessment, but their coupling in complex gene subnetworks renders a direct link to such scores difficult. An interesting observation was the absence of genes with high confidence of ASD risk in the DD network. This provides further support for the validity of our framework. Most genes had no information on their individual relevance to ASD risk, which emphasizes the relevance of exploring polygenic risk to differentiate complex diseases. Accordingly, we found clearly distinct patterns in both ASD and DD categories. The number of genes and links found in these networks represent a challenge in terms of biological interpretation, which led to the ontology-based approach.

The ASD GO Network and DD GO Network (Figure 10 and Figure 11, respectively) were found to be better suited for biological interpretation due to their network structure and the conceptual meanings that could be associated to nodes. These networks appear connected in one single component, which is a big difference from their diagnosis gene network counterparts. Importantly, this fact strongly suggests that behind the genetically variable nature of these complex disease exists a meaningful connectivity between the affected biologic mechanisms. This allowed us to identify key dopaminergic concepts which were differentially disrupted in each diagnosis. For example, in the diagnosis of ASD we identified the disruption of four key biological mechanisms related to dopamine-receptor binding, dopamine-metabolic processing, the regulation of dopaminergic neuron differentiation and the dopaminergic synapse. Moreover, these concepts appear strongly tied in network communities formed by specific domain disruptions related to these key concepts. This highly suggests that different subsets of ASD participants only had disruptions within one of these domains, i.e., only belong to one community trait.

However, given the network’s single-component pattern and the existence of such communities, it is also likely that different subsets of participants could share traits from more than one community, and in particular from those in the close neighborhood. For example, the dopaminergic-synapse community and the regulation of dopaminergic neuron differentiation were close neighbors. It is therefore likely that different combinations of disruptions in these communities may be behind some groups of ASD participants. Thus, this feature could be a major factor to understand the ASD heterogeneity derived from genetic alterations.

Additionally, we found that DD only had one community where several key dopaminergic concepts of different domains were disrupted. This is another clue about the neurobiological distinctions between ASD and DD, and could explain the different characteristics between these diagnostic categories.

Lastly, we transformed the information derived from network analysis to address a differential diagnosis classification problem (Table 4). Our best accuracy (85.18%) was only 2.02% behind the best classifier found in the studies mentioned earlier, which used EEG brain features [61]. Indeed, our study surpasses in accuracy the study using microarray data by about 11.51% [21].

Additionally, it used a larger sample size than the previous studies and showed consistency over the several ML models and metrics obtained. Here, we highlight the importance of the nature of the features used to train a classifier. For example, a classifier that relies only on gene-dosage information to decide may not be able to give an accurate response in the presence of a gene that the classifier was not trained on beforehand. On the other hand, if the classifier uses GO information it may be able to infer a correct response based on the previous examples, because it is not tied to a set of genes but instead to a set of biological processes, molecular functions and cellular components relevant to the problem.

Indeed, as we demonstrated throughout the study, GO information gives a less-variable and more-compact set of domain-oriented features. Thus, we consider this framework a solid option when developing knowledge-based systems addressing support decisions in ASD. One approach for developing support-decision tools in this field should be to consider collecting and weighting the response of an ensemble of classifiers [72] working with different types of data (ex.: genes, GO, brain data) to deliver a diagnostic-output response.

Furthermore, to extend this approach beyond the functions of the dopaminergic system in ASD, it could be interesting to address several ML classifiers to each system or pathway shown to be disrupted in ASD. It is likely that the performance of this approach depends on the type of differential diagnosis and classification being performed. Combining the response of distinct classifiers will probably increase the diagnostic accuracy of the response while, at the same time, analyzing the individual output of each classifier may provide insights about the systems disrupted in a given ASD case.

Importantly, ASD is a neurodevelopmental disorder strongly tied to behavioral aspects. This is an opportunity to improve our knowledge about gene–brain-behavioral relationships. In future studies we will explore this by building ML approaches to attempt predicting neurobehavioral features in ASD.

The clinical potential of the approach present here can be useful in two clinical domains, namely, diagnosis and intervention. The methodology used to generate vectors containing genetic information from multiple genes and several biologic mechanisms that describe the network of interactions between these and the disease and a machine learning algorithm could be integrated in a clinical setting in order to provide support to clinical decisions and streamline the clinical diagnosis process. For intervention, it could be a valuable tool used as a recommendation support system. For example, a network of interactions between diseases, genes and interventions could allow it to recommend an intervention based on genetic and disease similarities between participants. These solutions will benefit most approaches aiming to deliver personalized medicine services.

To conclude, we would like to discuss some study limitations. The type of CNV data used here are usually gathered from blood or saliva samples and reflect genetic variations of more than 50 k base pairs. This excludes potential interactions of genetic mutations from the analysis as well as any attempt to study DNA extracted from the local of interest, for example.

Additionally, the context (quantitative behavioral genetic studies) and the aim (to estimate the relative contribution of genetic variation in defining the variation in the trait of study across a population) in and for which these data are used output only CNVs of statistical evidence for genetic influence. Therefore, the interplay with CNVs with insignificant statistical evidence could not be studied as well as the interplay with common variants, which could potentially impact the study outcomes. This is a trade-off when addressing this issue with data from epidemiologic studies.

Furthermore, Gene x Environment interactions could not be included here. This is a major issue and a very challenging one to address. The expression of a gene and its rate is dependent of several factors: state of the nucleotide sequence, epigenetic marks, intracellular conditions, cellular signaling and time are all varying factors which together contribute to the final phenotypic expression at the biological level.

Yet, we uphold the methodology and the results obtained here, which were successful at: (1) identifying potentially major imbalanced genetic and biologic mechanics of dopaminergic synapses and their potential interactions in a large sample of participants with ASD or DD; and (2) extracting relevant biological meaning and diagnostic differences from a complex problem; which could be used to perform successful classification using biologic features extracted from complex networks and machine learning algorithms.

## Figures and Tables

**Figure 1 jpm-12-01579-f001:**
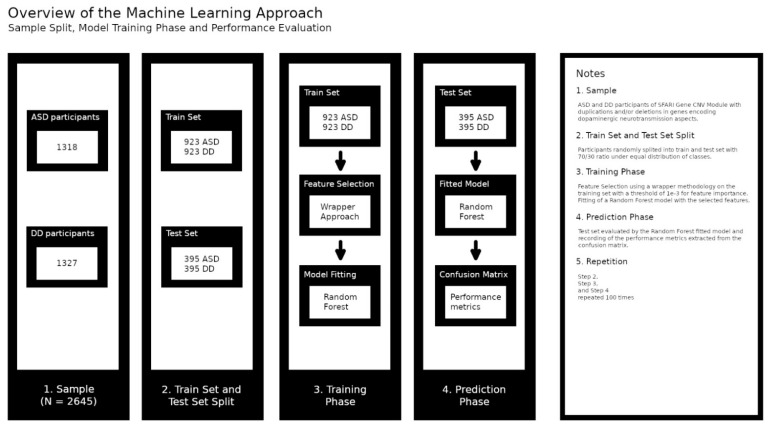
Overview of the Machine Learning approach. The sample was randomly split into a training and a testing set. A feature reduction using a wrapper approach operating with a Random Forest algorithm was performed on the training set. Next, the training set was used to train a Random Forest algorithm. The test set was then accessed by the Random Forest algorithm and the classifier performance was recorded. This process was repeated 100 times.

**Figure 2 jpm-12-01579-f002:**
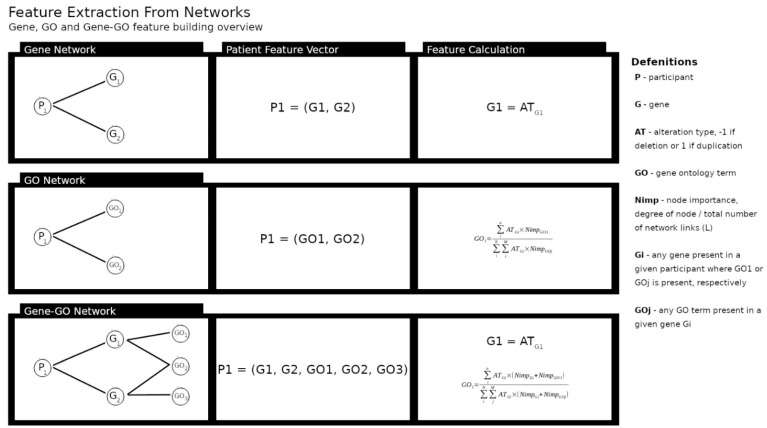
Example of feature extraction from the networks. In the Gene Network the vector of features is given by multiplying each gene node importance by the type of alteration (1 for duplications or −1 for deletions) that links a participant to a gene. In the GO Network the same principle was applied to build the features vector; when a GO term was shared by multiple genes its product was summed. In the Gene–GO Network a participant feature vector was built inserting the gene node importance and the GO node importance. However, in this network the GO term was influenced by the importance of the gene node it was linked to. This step allowed us to weight differently GO terms shared by multiple genes.

**Figure 3 jpm-12-01579-f003:**
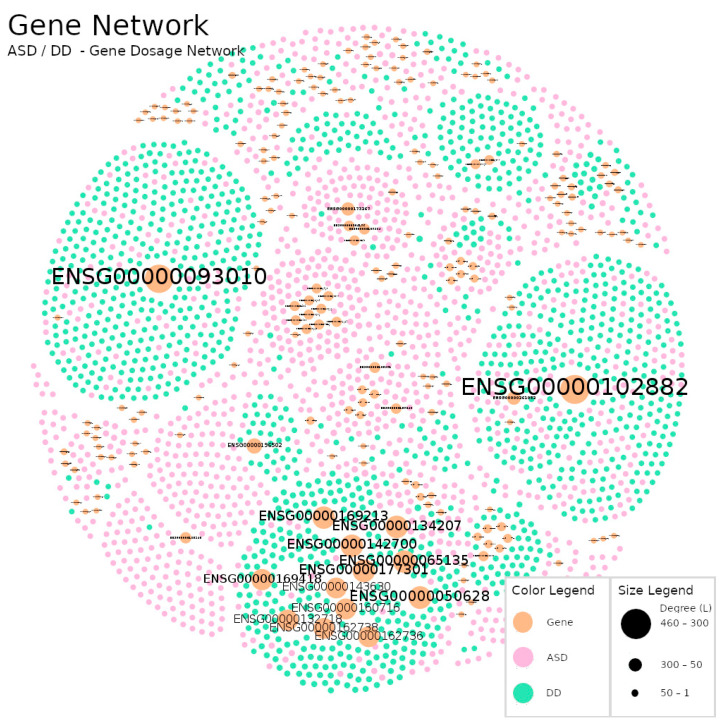
Gene Dosage Network: A network of participants with ASD (pink nodes) or DD (green nodes) diagnosis and their links to duplicated or deleted genes (orange nodes). The node labels represent a unique gene identification attributed by Ensembl (ENSG nodes). The size of a node reflects its own degree. In this network, ASD nodes represent 45.49% of the total number of the nodes (N on Table 2), DD 47.08% and genes 7.44%. Links were omitted for visualization purposes. Produced with Gephi software (version 0.9.2, created by Mathieu Bastian, Sebastien Heymann and Mathieu Jacomy, Paris, France), using the Fruchterman–Reingold layout (parameters: Area = 10,000, Gravity = 5, and Speed = 5).

**Figure 4 jpm-12-01579-f004:**
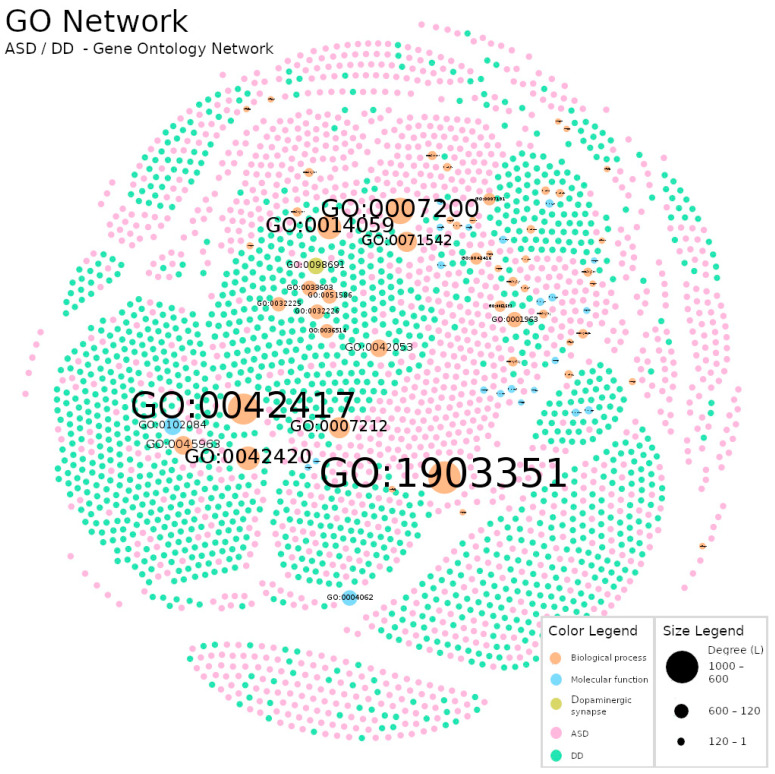
GO network. A network of participants with ASD (pink nodes) or DD (green nodes) diagnosis where the links to their genes were replaced by GO terms: biological processes (orange nodes), molecular functions (blue nodes) and the cellular component representing the dopaminergic synapse where gene-products perform actions (olive node). The node labels represent a unique term identification established by GO. The size of the node reflects its own degree. In this network, ASD nodes represent 48.43% of the total number of the nodes (N on Table 2), DD 48.84%, biological processes 1.88%, molecular functions 0.81% and the cellular component 0.04%. Links were omitted for visualization purposes. Produced with Gephi software (version 0.9.2, created by Mathieu Bastian, Sebastien Heymann and Mathieu Jacomy, Paris, France), using the Fruchterman–Reingold layout (parameters: Area = 10,000, Gravity = 5, and Speed = 5).

**Figure 5 jpm-12-01579-f005:**
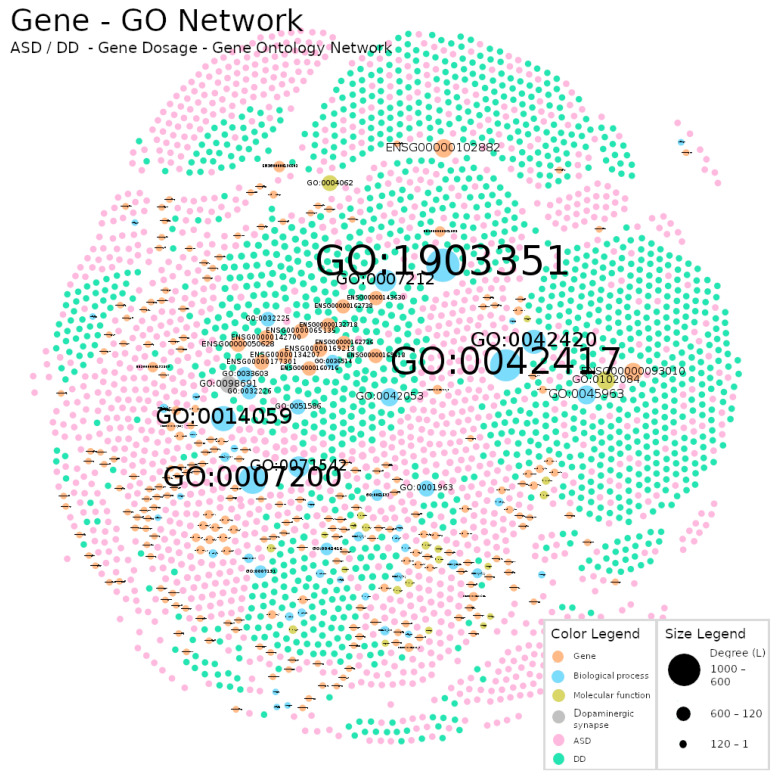
Gene Dosage–GO Network. A network of participants with ASD (pink nodes) or DD (green nodes) diagnosis, their links to their genes (orange nodes) and to GO terms: biological processes (blue nodes), molecular functions (olive nodes) and the cellular component (grey node). The gene labels refer to Ensembl-attributed unique gene identifiers, and GO term labels represent unique GO identifiers. The size of the node reflects its own degree. In this network, ASD nodes represent 44.65% of the total number of the nodes (N on Table 2), DD 44.95%, genes 7.89%, biological processes 1.73%, molecular functions 0.75% and the cellular component 0.03%. Links were omitted for visualization purposes. Produced with Gephi software (version 0.9.2, created by Mathieu Bastian, Sebastien Heymann and Mathieu Jacomy, Paris, France), using the Fruchterman–Reingold layout (parameters: Area = 10,000, Gravity = 5, and Speed = 5).

**Figure 6 jpm-12-01579-f006:**
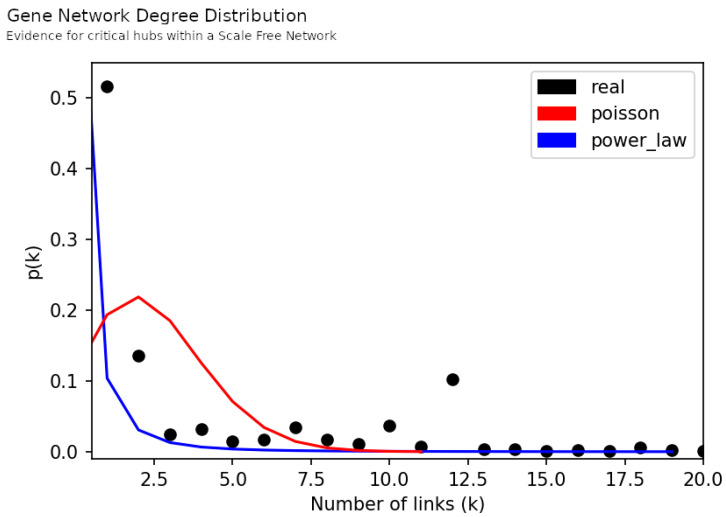
Evidence for critical hubs within a scale-free network as revealed by the Degree Distribution of the Gene Dosage Network (black dots), and the Poisson distribution (red curve) and the power-law distribution (blue curve) using the same number of nodes and links as the Gene Dosage Network. On this figure the probability threshold was set to 0.0001 (y axis) and the number of links set to a maximum of 20 (x axis). The maximum number of links is reported on Table 3.

**Figure 7 jpm-12-01579-f007:**
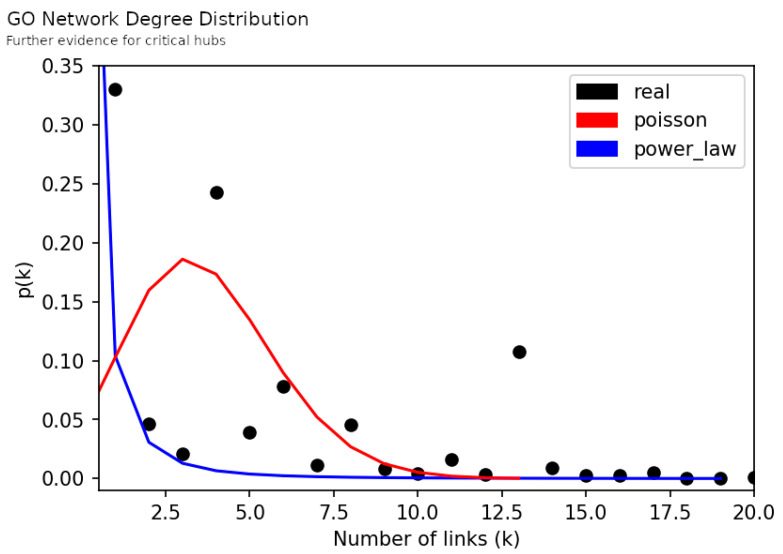
Further evidence for critical hubs as revealed by the Degree Distribution of GO Network (black dots), and the Poisson distribution (red curve) and the power-law distribution (blue curve) using the same number of nodes and links as the GO Network. On this figure the probability threshold was set to 0.0001 (y axis) and the number of links set to a maximum of 20 (x axis). The maximum number of links is reported on Table 3.

**Figure 8 jpm-12-01579-f008:**
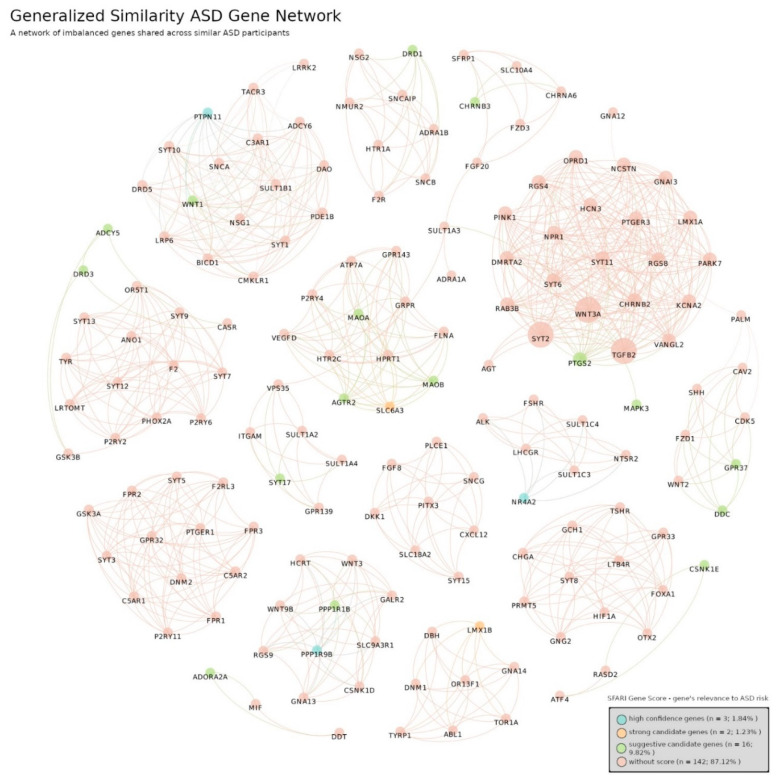
ASD Gene Network. A network of imbalanced genes shared across similar ASD participants. The network is composed of 163 different genes spread in 21 communities. It is a relatively disconnected network of 17 connected components where each component represents a different group of ASD participants. Number of links: 832, density: 0.063 and modularity: 0.923. The colors represent the individual gene’s strength of ASD risk: blue represent genes of high evidence (3 genes, 1.84%), orange are strong candidate genes (2 genes, 1.23%), green represent suggestive candidate genes (16 genes, 9.82%) and light red marks genes without information of individual risk (142, 87.12%). This risk is calculated by SFARI Gene in its Gene Score Module. The links between gene nodes were obtained using the generalized similarity algorithm (version 0.9.2, created by Mathieu Bastian, Sebastien Heymann and Mathieu Jacomy, Paris, France) [45] and the image was produced with Gephi software, using the Fruchterman–Reingold layout (parameters: Area = 10,000, Gravity = 5, and Speed = 5).

**Figure 9 jpm-12-01579-f009:**
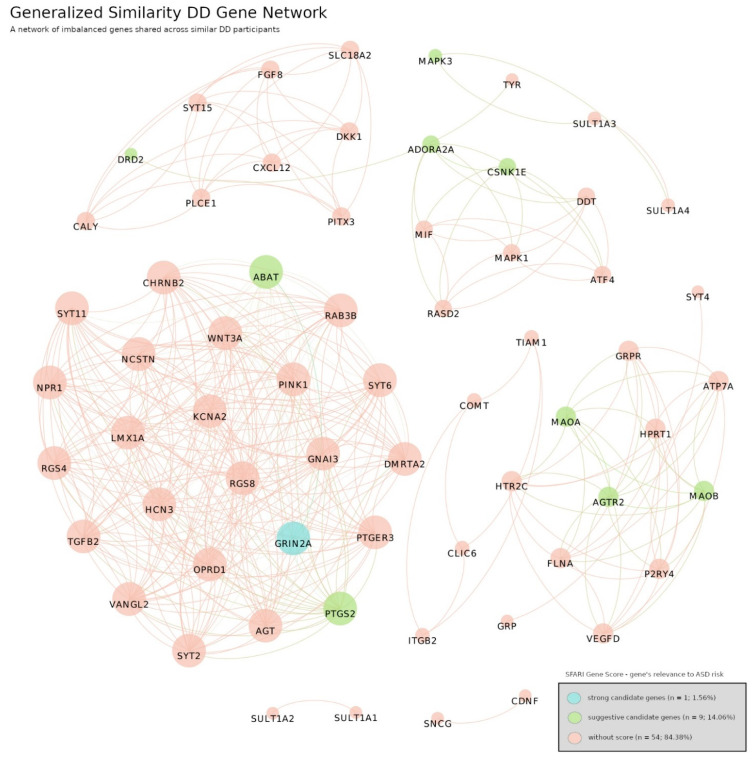
DD Gene Network. A network of imbalanced genes shared across similar DD participants. The network is composed of 64 different genes spread in 11 communities. It is a disconnected network of 10 connected components where each component represents a different group of DD participants. Number of links: 383, density: 0.19 and modularity: 0.604. The colors represent the individual gene’s strength of ASD risk: an absence of genes of high evidence (0 genes, 0.0%), blue is the only strong candidate gene (1 gene, 1.56%), green represent suggestive candidate genes (9 genes, 14.06%) and light red marks genes without information of individual risk (142, 87.12%). This risk is calculated by SFARI Gene in its Gene Score Module. The links between gene nodes were obtained using the generalized similarity algorithm [45] and the image was produced with Gephi software (version 0.9.2, created by Mathieu Bastian, Sebastien Heymann and Mathieu Jacomy, Paris, France), using the Fruchterman–Reingold layout (parameters: Area = 10,000, Gravity = 5, and Speed = 5).

**Figure 10 jpm-12-01579-f010:**
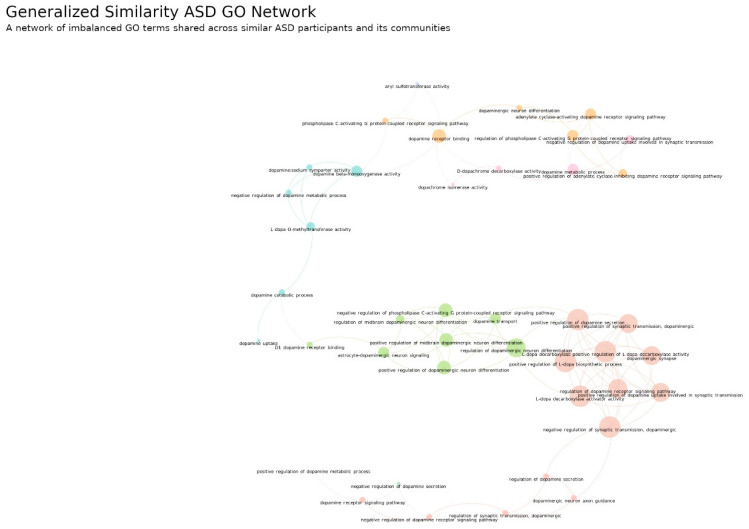
ASD GO Network. A network of disrupted GO terms shared across similar ASD participants. The network is composed by 41 different GO terms spread in 9 differently colored communities. It is a connected network where each community represents a different set of GO terms dysregulated per subset of ASD participants. The interconnected communities are related to ASD participant subsets that share traits of more than one community. Number of links: 100, density: 0.122 and modularity: 0.71. The links between nodes were obtained using the generalized similarity algorithm [45] and the image was produced with Gephi software (version 0.9.2, created by Mathieu Bastian, Sebastien Heymann and Mathieu Jacomy, Paris, France), using the Fruchterman–Reingold layout (parameters: Area = 10,000, Gravity = 5, and Speed = 5).

**Figure 11 jpm-12-01579-f011:**
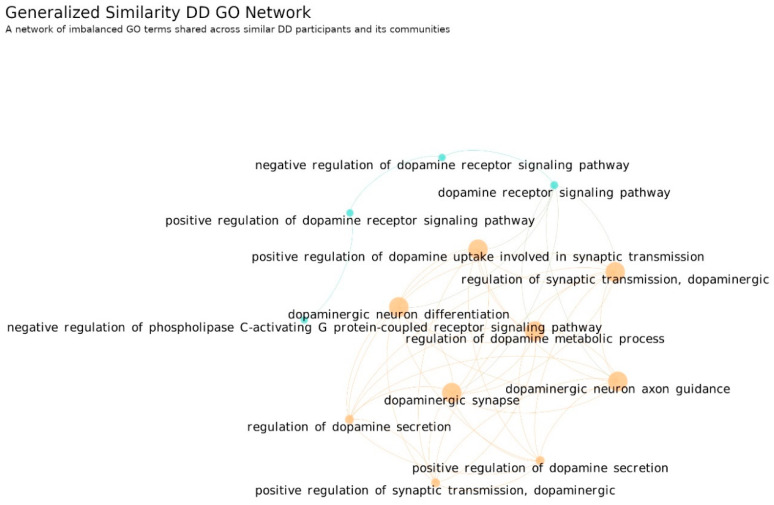
DD GO Network. A network of disrupted GO terms shared across similar DD participants. The network is composed by 13 different GO terms spread in two differently colored communities. It is a connected network where each community represents a different set of GO terms dysregulated per subset of ASD participants. Number of links: 45, density: 0.572 and modularity: 0.097. Since its modularity is very low it is possible to assume only one community which represents most of the disruptions found in the DD participants. The links between nodes were obtained using the generalized similarity algorithm [45] and the image was produced with Gephi software (version 0.9.2, created by Mathieu Bastian, Sebastien Heymann and Mathieu Jacomy, Paris, France), using the Fruchterman–Reingold layout (parameters: Area = 10,000, Gravity = 5, and Speed = 5).

**Table 1 jpm-12-01579-t001:** Distribution of Participant Diagnosis: Diagnosis and the number of participants (N) identified with duplicated or deleted genes containing information related with dopaminergic aspects in SFARI Gene CNV Module. The sex of the participants is reported in the Sex column as males (M), females (F) or not specified (NS). Only participants with a single diagnostic category were considered. * denotes participants investigated in this study.

Diagnosis	Total (N)	Sex(M/F/NS)
Developmental Delay (DD)	1327 *	81/67/1179
Autism Spectrum Disorder (ASD)	1318 *	699/134/485
Schizophrenia	50	18/10/22
Intellectual Delay (ID)	41	20/18/3
Middle Cerebral Artery Syndrome (MCA)	23	8/11/4
Epilepsy	17	11/6/0
Childhood Apraxia of Speech (CAS)	13	6/3/4
Polymicrogyria (PMG)	10	0/0/10
Attention Deficit and Hyperactivity Disorder (ADHD)	6	4/0/2
Bipolar Disorder	3	1/2/0
Schizoaffective Disorder	3	0/3/0
Borderline Personality Disorder (BPD)	1	0/1/0
Congenital Heart Disease (CHD)	1	0/1/0
Microcephaly	1	0/1/0
Angelman Syndrome	1	0/1/0

**Table 2 jpm-12-01579-t002:** Network Centrality Measures: the total number of nodes (N), the total number links (L), the average degree for unidirectional networks (<k>), the density, the diameter and the radius of each network.

Network	N	L	<k>	Density	Diameter	Radius
Gene Dosage	2770	9387	3.389	0.001	16	8
GO	2719	12669	4.659	0.002	7	4
Gene Dosage–GO	2952	22568	7.645	0.003	7	4

**Table 3 jpm-12-01579-t003:** Networks’ maximum degree nodes (hubs): the node, their name, node degree (k), and the average degree of their neighbors of the Gene Dosage and of the GO Networks.

Network	Hub	Name	k	Average Neighbors Degree
Gene Dosage	ENSG00000102882	*MAPK3*	463	1.6
	ENSG00000093010	*COMT*	450	1.2
	ENSG00000050628	*PTGER3*	339	11.8
GO	GO:1903351	Cellular response to dopamine	1029	6.3
	GO:0042417	Dopamine metabolic process	968	7.5
	GO:0007200	Phospholipase C-activating G protein-coupled receptor signaling pathway	788	8.9

**Table 4 jpm-12-01579-t004:** Machine Learning Results of differential diagnosis classification between ASD and DD.

Dataset		Tn	Fp	Fn	Tp	N Features	Acc Train (%)	Acc Test(%)	AUC	DD Precision (%)	DD Recall (%)	ASD Precision (%)	ASD Recall (%)	DD f1 Score (%)	ASD f1 Score(%)
Gene Dosage	mean	369.6	25.4	91.7	303.3	117.4	88.59	85.18	0.85	80.15	93.57	92.30	76.79	86.33	83.82
	sd	5.3	5.3	7.7	7.7	3.9	0.41	1.11	0.01	1.35	1.35	1.47	1.96	0.98	1.31
GO	mean	368.0	27.0	105.5	289.5	62.0	86.25	83.22	0.83	77.73	93.15	91.50	73.28	84.73	81.36
	sd	6.6	6.6	7.4	7.4	1.5	0.41	1.09	0.01	1.18	1.68	1.81	1.87	0.99	1.28
Gene Dosage–GO	mean	371.8	23.2	94.3	300.7	58.6	88.59	85.13	0.85	79.79	94.13	92.87	76.13	86.36	83.66
	sd	5.0	5.0	7.2	7.2	1.6	0.37	1.06	0.01	1.24	1.27	1.42	1.83	0.94	1.25

The mean and the standard deviation of the measures obtained from 100 repetitions in the training and the testing of a Random Forest classifier for each dataset. (tn: true negative, fp: false positive, fn: false negative, tp: true positive, N features: the total number of features used to train the classifier, Acc train: accuracy of the classifier in the train set, Acc test: accuracy of the classifier in the test set, AUC: area under the roc curve, DD precision: precision measured for DD class, DD recall: recall of DD class, ASD precision: precision of ASD class, ASD recall: recall of ASD class, DD f1 score: F1 score of DD class, ASD f1 score: F1 score of ASD class. Positive class: ASD; Negative class: DD; Train size: 1846 participants (923 ASD/923 DD); Test Size: 790 participants (395 ASD/395 DD)).

## Data Availability

The raw data as well as the data supporting reported results can be found in this repository: A.S., F.C., J.B.M., M.C.-B. Study repository: A relational database of SFARI Gene CNVs data integrated with associated genes and GO terms for the study of genetics in neurodevelopmental disorders—Autism Imaging Genetics Dataverse (Internet). 2022 (cited 8 March 2022). Available online: https://dataverse.harvard.edu/dataset.xhtml?persistentId=doi:10.7910/DVN/HO1JLJ, accessed on 8 March 2022.

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
