# Peer review of "Dopaminergic Gene Dosage Reveals Distinct Biological Partitions between Autism and Developmental Delay as Revealed by Complex Network Analysis and Machine Learning Approaches"

_jpm, 2022, doi:10.3390/jpm12101579_

Round 1

Reviewer 1 Report

The study reported interesting insights on the neurobiology of ASD and DD. Using complex network analysis, the authors investigated the relations between phenotypic profiles, gene dosage and GO terms related to dopaminergic neurotransmission. 

The resulted data were fed into a machine learning classification to differentiate between ASD and DD. This data may be used for diagnosis based on polygenetic profiles in clinical settings.

Introduction 

1) The authors should explicitly mention the other fundamental aspect associated with ASD: few and stereotyped interest

2) "Research on putative causal aspects that conduct to the different manifestations of ASD remains a priority"

The term "manifestation" is unclear: does it refer to different symptoms present in the same individuals or different clinical conditions?

3) The authors should explicitly mention that in the last version of DSM (DSM 5), there is only one macro clinical condition "ASD". Clinicians could specify only the severity of the disease, but there is only "ASD" diagnosis (while in the previous version of DSM there were more diagnostic profiles)

4) "A 2019 meta-analysis on the subject [20], observed a discrepancy in the diagnostic prediction accuracy between 60% and 98% across 57 different studies."

The authors should report the methods used in this meta-analysis.

Materials and Methods 

5) "Participants with diagnostic comorbidities were excluded"

It is unclear if the authors selected only ASD participants with average IQ and lack of language difficulties. They should clarify which patients are included in their analyses

6) "The target classification variable was the differential diagnosis of participants"

Which test o data are used for this diagnosis?

Results

7) Table 1: Considering the different clinical manifestations between males and females, the authors should specify the gender of their sample. 

Moreover, if the authors split their ML analysis between females and males, is there some different classification accuracy?

Discussion

8) "Our approach is novel and allowed to address issues, such as heterogeneity, which has been largely pointed out by previous studies."

Is it correct to use the term "heterogeneity", considering that the analysis is focused only on ASD participants with comorbidities?

9) The discussion needs to be more focused on the potentiality of this approach in the clinical setting, for example, in terms of fast diagnosis and intervention. 

Author Response

Response to REVIEWER-1

1) The authors should explicitly mention the other fundamental aspect associated with ASD: few and stereotyped interest

RESPONSE: We now provide a more comprehensive description that Autism Spectrum Disorder (ASD) is a neurodevelopmental disorder characterized by social and communication impairments and restrictive and repetitive behaviours and interests. We now cite articles from our group emphasizing this aspect.

2) "Research on putative causal aspects that conduct to the different manifestations of ASD remains a priority"

The term "manifestation" is unclear: does it refer to different symptoms present in the same individuals or different clinical conditions?

RESPONSE: We mean different symptoms for the same individuals.

3) The authors should explicitly mention that in the last version of DSM (DSM 5), there is only one macro clinical condition "ASD". Clinicians could specify only the severity of the disease, but there is only "ASD" diagnosis (while in the previous version of DSM there were more diagnostic profiles)

RESPONSE:  we now clarify that the attribution of the ASD diagnosis is assessed accordingly with the guidelines pre-sent in the last version of DSM (DSM-5). It admits only one macro clinical condition, ASD, and allows clinicians to specify severity of different symptoms (which differs from previous versions of DSM where different diagnostic profiles existed to differentiate different symptom severity).

4) "A 2019 meta-analysis on the subject [20], observed a discrepancy in the diagnostic prediction accuracy between 60% and 98% across 57 different studies."

The authors should report the methods used in this meta-analysis.

RESPONSE: The authors conducted a literature search on all studies that attempted to predict ASD clinical diagnosis status either cross-sectionally or longitudinally on the basis of biological, cognitive and/or behavioural features, implementing a case-control design and with only out-of-sample predictions (i.e., a predictive model was trained on one part of the data and tested on another). This is now clarified in the manuscript.

Materials and Methods

5) "Participants with diagnostic comorbidities were excluded"

It is unclear if the authors selected only ASD participants with average IQ and lack of language difficulties. They should clarify which patients are included in their analyses

RESPONSE We now clarify this part of the manuscript. We did only include in the analysis participants with a single formal clinical diagnosis, based on the recognised diagnostic criteria irrespective of other features.

6) "The target classification variable was the differential diagnosis of participants"

Which test o data are used for this diagnosis?

RESPONSE. This was according to the clinical criteria used in the SFARI initiative.

Results

7) Table 1: Considering the different clinical manifestations between males and females, the authors should specify the gender of their sample.

RESPONSE: The sex of the participants is now specified in table 1. M-male, F-female and, NS-not specified

Moreover, if the authors split their ML analysis between females and males, is there some different classification accuracy?

We did not split the ML (machine learning) analysis between males and females for two main reasons:

  • the number of samples without a specified sex. There are 1664 samples without a specified sex (1179-DD, 485-ASD).
  • the number of samples without a specified sex between ASD and DD diagnosis is very imbalanced. In ASD there are 833 participants (M-699, F-134) with sex information whereas in DD only 148 participants (M-81, F-67) have this information specified.

Given these two facts we thought that a sex analysis will put in cause one of our goals with this analysis, study genetic variability in developmental disorders using a polygenetic model with a large sample. Removing 1664 samples will have reduced the amount of information available to build the predictive models, resulting in a cut of genes and Gene Ontology terms studied. This action would probably impact the ML results since the number of samples and the number of features available will differ between ML analysis. Consequently, it would be difficult specify if the difference observed was a result of splitting between sex or a result from a cut in the sample size and number of features. Thus, we made the option to make the ML analysis with the largest sample we could find without split between sex.

However, we recognise that the sex is an important analysis factor in the study of neurodevelopmental disorders, particularly in ASD. This was something that we have in mind when we thought about the strategy to map the participants genetic information into vectors for ML analysis. Our methodology to build the participants vectors can describe genetic differences between participants. Given the fact that sex is a chromosomic feature it is captured in the participant gene vectors, if any of those genes are duplicated or deleted in sex chromosomes. Importantly, the methodology used to build the participants genetic vectors can be extended beyond gene duplications or deletions. Here, we only have access to this type of genetic variations, but with other types of genetic information it is possible to increase the information captured in a vector, including sex features related to genes present in sex chromosomes. Such vectors, for example, could allow to build ML algorithms that recognise sex differences without the need to split between sex.

Discussion

8) "Our approach is novel and allowed to address issues, such as heterogeneity, which has been largely pointed out by previous studies."

Is it correct to use the term "heterogeneity", considering that the analysis is focused only on ASD participants with comorbidities?

RESPONSE We agree that some clarification is needed here. We meant genetic variability

9) The discussion needs to be more focused on the potentiality of this approach in the clinical setting, for example, in terms of fast diagnosis and intervention.

 RESPONSE The clinical potential of the approach present here can indeed be useful in two clinical domains, namely, diagnosis and intervention. The methodology used to generate vectors containing genetic information from multiple genes and several biologic mechanisms that describe the network of interactions between these and the disease and a machine learning algorithm could be integrated in a clinical setting in order to provide support to clinical decisions and streamline the clinical diagnosis process. For intervention, it could be a valuable tool used as a recommendation support system. For example, a network of interactions between diseases, genes, and interventions could allow to recommend an intervention based on genetic and diseases similarities between participants. These solutions will benefit most approaches aiming to deliver personalized medicine services. These points are now addressed in the manuscript.

Reviewer 2 Report

The manuscript entitled "Dopaminergic gene dosage reveals distinct biological partitions between autism and developmental delay as revealed by complex network analysis and machine learning approaches" by Santos et al. describes the network differences between ASD and DD using a large number of test samples. The authors have utilized the resources well and sed complex network analysis using phenotypic profiles, gene dosage, and gene ontology for this study.  

A few minor corrections are required to improve the manuscript.

  1. Figure 3, the ENSG-related description, is added in the text, but its corresponding figure description is missing. Adding this information to the figure legend would be beneficial for the audience to understand the data well. 
  2. Figure 10 is blurred, and not able to see anything. 
  3. The participant's ASD degree levels (high, medium, or low) are missing. 
  4. Male and female ratios of ASD and DD are also missing in the methods or results section. 
  5. Line 49 appears not described well and should be described more clearly. 
  6. Line 316, remove the comma for the total percentage (2.73%).

Author Response

Reply to REVIEWER-2

1) Figure 3, the ENSG-related description, is added in the text, but its corresponding figure description is missing. Adding this information to the figure legend would be beneficial for the audience to understand the data well.

RESPONSE: done

2) Figure 10 is blurred, and not able to see anything.

RESPONSE: -> We apologize about poor quality of the figure 10. We had difficulties in display the Figure 10 properly in the manuscript due to the algorithm that layout the nodes and the node labels in the image and the export function provided by the Gephi software. We tried several approaches, but we were not able to obtain a better figure to display directly in the word file manuscript.

However, we attached a separate folder with all figures used in the manuscript with 400 dpi. In this folder, the figure 10 can be open in an image viewer and its labels are readable when zoomed in without quality loss. We hope that this folder could be included in the final version to mitigate this problem.

3) The participant's ASD degree levels (high, medium, or low) are missing.

RESPONSE:  The ASD degree levels were not available to us in the data source used for this work (SFARI Gene data)

4) Male and female ratios of ASD and DD are also missing in the methods or results section.

RESPONSE  Male and female ratios were added to the manuscript in methods section (2.1. Data source and participant selection) and in results section (Tabel 1)

5) Line 49 appears not described well and should be described more clearly.

RESPONSE We now rewrote this line: Autism Spectrum Disorder (ASD) is a neurodevelopmental disorder characterized by social and communication impairments and restrictive and repetitive behaviours and interests, yet its aetiology and neurobiological mechanisms are still poorly understood

6) Line 316, remove the comma for the total percentage (2.73%).

RESPONSE -> Updated to 2.73%

Round 2

Reviewer 1 Report

The paper is improved and I think it can be published.